# OpenTeleView: An Open 3D Teleconferencing Research Platform

Category: Research

## ABSTRACT

Recent demonstrations of 3D telepresence provide a glimpse into a future where 2D video communication is replaced with photo-realistic virtual avatars rendered on 3D displays. However, the existing technology demonstrations typically run on expensive dedicated devices that require the calibration of multiple cameras by experts and the underlying reconstruction, compression, transmission, and rendering methods remain proprietary. We describe our open platform for real-time end-to-end 3D teleconferencing using commodity hardware coupled with a modular software structure for inserting advanced computer vision algorithms supporting research and development. We demonstrate the utility of our modular end-to-end approach by integrating state-of-the art modules and improving them based on an analysis of current bottlenecks targeting low-latency processing. We include a baseline implementation supporting real-time 3D teleconferencing that provides a new benchmark for evaluation of current and future algorithms. We demonstrate the practicality of our approach with a baseline, a 3D teleconferencing system running at 25 frames per second with 172 ms latency on consumer GPUs that applies to a single RGB camera input and various 3D display technologies. Our 3D teleconferencing platform is open source, which paves the way for computer vision, computer graphics and HCI research to continue innovating together to make 3D teleconferencing the telecommunication standard.

## 1 INTRODUCTION

With the dramatically accelerated shift to online meetings from the impact of the COVID-19 pandemic, there has been a resurgence in the need of new teleconferencing technology that creates a more real and in-person experience. One major challenge is to make teleconferencing have feeling of presence including eye contact and situational awareness of each person's real-world space, such that pointing, and gesture are coordinated. Hence, more research effort is appearing for teleconferencing that allows the user to appear in 3D and maintain direct eye contact with multiple speakers to enhance the overall communication experience and improve the information transmission efficiency [25]. Virtual Reality (VR) and Augmented Reality (AR) are the two main trends to create 3D experiences in recent years. These trends use three different types of hardware: headsets (HMDs) that connect to your PC, 2D semi-transparent displays like Google Glasses, and standalone 3D display devices. These displays support view-dependent rendering such as used in Fish Tank Virtual Reality (FTVR) that creates an effective method to support presence with stereo and motion parallax depth cues. However, these systems require rendering a person's likeness from different viewpoints which is not available without using some mechanism to capture and transmit the users' 3D characteristics. A number of proprietary systems have been proposed to achieve this goal, e.g., Google Starline project [26], Microsoft Holoportation [35], and [36], but each has either closed systems or large scale proprietary or prohibitively expensive hardware. Likewise, they are unavailable for researchers to perform perceptual evaluation to determine how well they achieve a sense of presence. Furthermore, the complex infrastructure to test proposed new research algorithms for supporting different aspects of the 3D teleconferencing pipeline is not readily accessible; thus, researcher results are typically reported in isolation without the opportunity to stress test it within the ecosystem of an end-to-end system. Our contribution fills this missing piece.

We describe *OpenTeleView* (actual name hidden for review) plat-

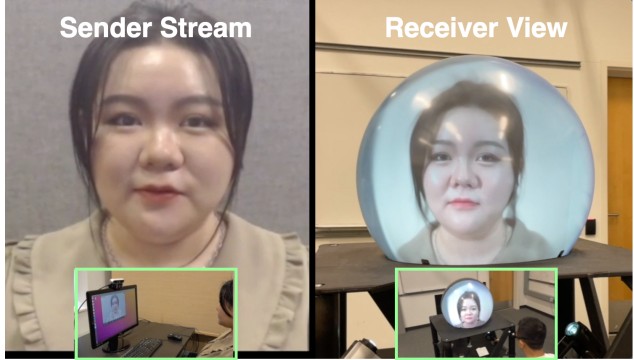

Figure 1: *OpenTeleView* modular End-to-End 3D teleconferencing in action. The Sender side camera captured image (left) is encoded to a neural 3D model. Its parameters are sent to the receiver side where a photo-realistic view-dependent rendering is shown on the Receiver's 3D display (right). Being modular, research results on different encoders can be substituted for analysis and comparison on real-world 3D teleconferencing experiences.

form that provides an end-to-end platform that supports researchers to include specific contributions to different parts of the pipeline in a 3D teleconferencing system. Within the platform, each component's performance can be tested within a perceptually suitable 3D teleconferencing system for benchmarking and optimization. We provide the end-to-end system that uses off the shelf (OTS) components along with our own adaptations of existing algorithmic approaches in the literature to demonstrate: a) an accessible, low-cost, replicable end-to-end 3D teleconferencing system with the latest advances in research included as a benchmark; b) interface descriptions that provide connections for research as well as the needed scaffolding to enable end-to-end functional and perceptual performance testing; c) a modular interface for researchers to connect to common development platforms like PyTorch and Unity; and, d) a high-resolution offline recording at 60 fps with novel-view ground truth to establish a public benchmark for 3D teleconferencing quality. Figure 1 shows an example of a user talking while showing her 3D image at the receiver's view-dependent display.

We provide results from experiments with the baseline implementation and variations to demonstrate how the platform can be used to help identify and optimize different types of algorithmic bottlenecks. Our implementation has an end-to-end latency of 172 ms with a sustained frame rate on average of 25 frames per second (FPS) providing an excellent reference point for innovative algorithms to be tested against. Besides as an algorithmic research platform, the technical performance is suitable for qualitative perceptual testing allowing different modules to be compared with each other in real-world user testing.

## 2 RELATED WORK

Research in teleconferencing has moved from 2D video to 3D. While significant research has gone into developing algorithms to make these systems feasible, we focus on the systems as a whole.

## 2.1 Talking Head Models

Parametric head models [2, 28] are widely used in face generation [16, 42] and reenactment [46–48]. These parametric models consume a low dimension vector that drives avatars to control the subjects. Following this line of work, we leverage the parametric model FLAME [28] in our baseline implementation and surround it with communication and rendering modules.

## 2.2 Neural Rendering

Different from traditional rendering methods [23], neural rendering does not necessarily need the explicit mesh and texture. It can be achieved by implicit neural representation [32], and Generative Adversarial Networks [24]. However, they usually focus on image quality for novel view synthesis [33,34] and object editing [7,14,43], both of which rely on very deep neural networks that only run at low frame rates. We utilize a parametric mesh model with the deferred neural rendering method [44, 45], aiming at high-resolution high-fidelity face synthesis at high frame rates and extend it to work alongside the other modules to form a complete teleconferencing system.

## 2.3 3D Teleconferencing

Gibbs et al. design a room-scale system which uses a single camera, a view tracking system, and IR emitter to render perspectively correct mono or stereo images on a wall-sized display [17]. Following that, [22] leverage a fast-rotating, convex mirror as a 3D display along with a high-speed projector to display a 3D image of a user. [29, 54] design a fully GPU-accelerated data processing and rendering pipeline and use a set of Microsoft Kinect color-plus-depth cameras to allow head-tracked stereo views to be rendered for a parallax barrier autostereoscopic display. [9, 36] design a room-scale telepresence setup which uses an array of color and depth cameras, and displays in two locations to synthesize images of users in both rooms with correct eye gaze. [25] use a single Microsoft Kinect depth camera and an RGB camera to render users from novel views without the need of a large camera array. This rendering is then shown on a 3D display over a 3D background. [53] use an array of IR cameras and lasers, RGB and Microsoft Kinect depth cameras to develop a system for three-person teleconferencing with proper eye gazes. Another line of work uses avatars or figures [6] as surrogates that circumvents the challenge of rendering a virtual avatar. More recently, [27] developed an end-to-end system which utilizes an array of cameras (IR, RGB, and tracking) and an autostereoscopic display among other contributions to enable face-to-face teleconferencing better than 2D alternatives. [30] uses a depth camera, and its 'inpainting' only supports moderate view changes. [52] and [31] are 2D, not capable of novel view synthesis. [50] could replace our FLAME-based encoder-decoder, but is not open source and the runtime is not stated. However, all of the recent live systems are proprietary and there is no publicly available offline benchmark.

## 3 END-TO-END PIPELINE

The challenge of 3D teleconferencing is finding compatible modules and connecting them to efficiently infer, transmit, and render a realistic 3D head model so that convincing 3D motion parallax and stereo depth cues are maintained as if the *Sender* appears at the *Receiver*'s location [57]. Figure 2 illustrates the main components of our *OpenTeleView* platform, with the *Sender/Receiver* hardware configuration, *Encoder* and *Decoder*, Persistent Data Storage (PDS) and communication module. The diagram shows the data flow from a *Sender* to a *Receiver* which would be duplicated for the bi-directional system; though they may have different camera and display configurations. The heart of the research for end-to-end 3D teleconferencing are the matched *Encoder* and *Decoder* for the encoding/compression of the input video signal and the subsequent decoding and view dependent rendering.

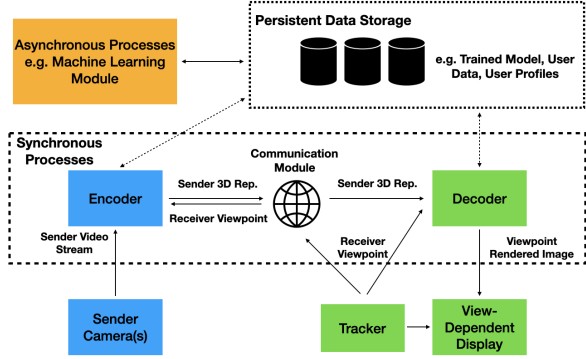

Figure 2: The main components needed by our *OpenTeleView* platform to define a 3D teleconferencing system are: 1. *Sender/Receiver* hardware, 2. *Encoder*, 3. *Decoder*, and 4. Persistent Data Storage needs. Our platform provides network scaffolding and communication interfaces, including optional access to the *Receiver*'s tracked position by the *Encoder* and *Decoder*, to support a range of end-to-end 3D teleconferencing research for performance testing, analysis and comparison.

We structured the *OpenTeleView* system to capture the main components that are necessary for an end-to-end 3D teleconferencing system and designed it to be modular, with the expectation that researchers will be able to add their own hardware assumptions with associated encoding and decoding approaches to strike different tradeoffs between quality and resources, e.g., for real-world perception testing as well as measurements of efficiency and quality of service. We provide sufficient scaffolding to accommodate a range of hardware assumptions, such as different display types, camera inputs and tracking technologies for rendering while providing software interfaces for supporting encoders and decoders doing frame-by-frame processing but also have access to persistent memory, accessed at start up when a connection is made between *Sender* and *Receiver* to exchange pre-trained models.

The communication infrastructure provides interfaces for inter-process communication to support modules to be run on different computers as well be written in different languages appropriate for the research.

We provide a baseline implementation with the *OpenTeleView* platform using a pre-trained head model and a neural render trained on *Sender* video data collected offline. Figure 3 shows the different components, each explained in detail in the subsequent sections. The *Encoder* generates a small set of 3D head parameters of the *Sender* that is sent to the *Decoder*. The head parameters capture enough 3D content so that the *Decoder* can recreate the head of the user along with a neural renderer trained on the *Sender*'s data that provides a photo-realistic, view-dependent render that can appear on the *Receiver*'s display. The neural renderer can continue rendering different view-points as the *Receiver* moves around their display as needed. To represent the *Sender* 3D head parameters, we use the FLAME [28] model because it is low-dimensional and more expressive than other representations, e.g., FaceWarehouse model [4] and Basel Face Model [37]. It is easy to fit to data and commonly used by many algorithms (e.g. RingNet [41]; DECA [13]; CoMA [38]).

FLAME's head representation include geometry parameters Since FLAME does not have an appearance model, like previous method [13], we adapt the Basel Face Model [37] to be compatible with FLAME to give albedo parameters $\alpha \in R^{50}$. Together, the *Encoder* (see Figure 3.3) computes these head parameters for every frame of the *Sender* and transmits these along with camera matrix $c$ and lighting parameters $l$ to the *Decoder*. The *Decoder*

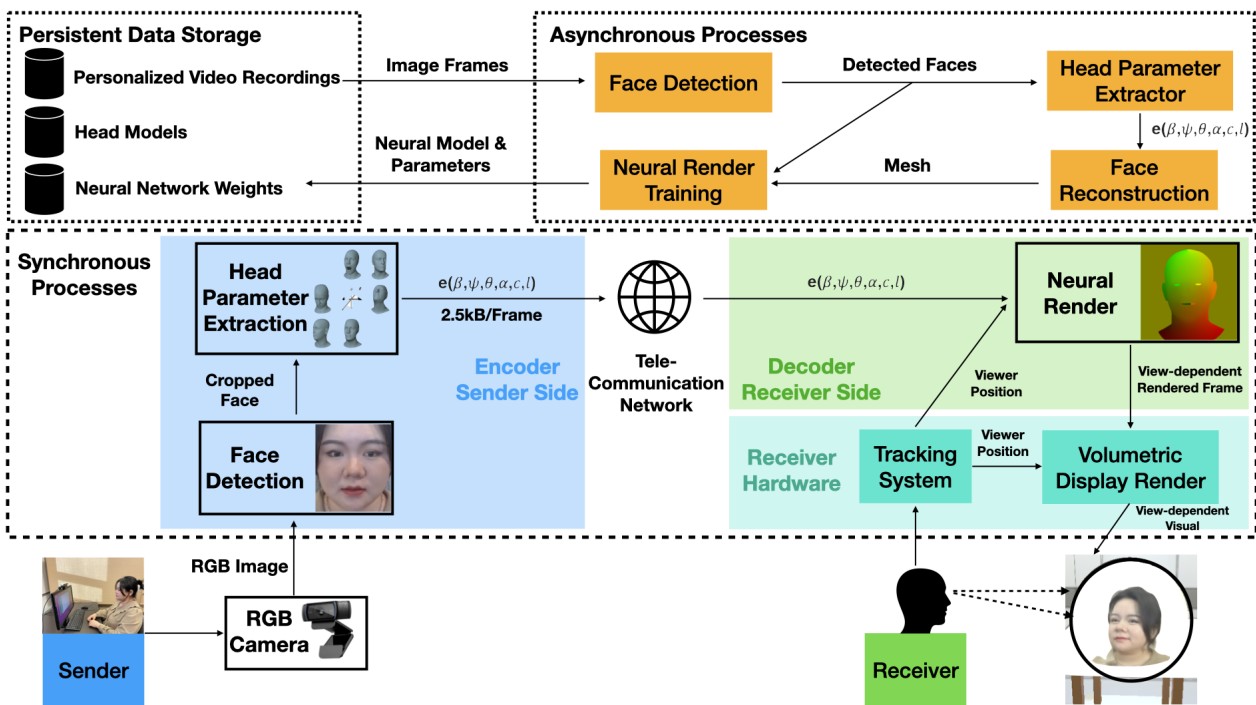

Figure 3: Baseline 3D Teleconferencing Architecture: *Encoder* and *Decoder* use a compact (2.5kB/frame) 3D head model that represents the *Sender*'s head using shape ($\beta$), expression ($\psi$), pose ($\theta$), albedo ($\alpha$), camera matrix ($c$), and lighting parameters ($l$). These are computed every frame by the *Encoder*, transmitted to the *Decoder* and decoded from a single RGB image. Using the *Receiver*'s viewpoint, the neural renderer renders a view-dependent photo-realistic image of the *Sender* on the *Receiver*'s 3D display.

(see Figure 3.4) then used them to reconstruct the 3D head model of the *Sender*. The neural renderer then maps the 3D head model to a photo-realistic version of the *Sender*, though, from the viewpoint of the *Receiver*.

3D head models and their rendering is an active research area for 3D teleconferencing, thus, our *OpenTeleView* platform makes it easy to analyse different approaches relative to each other in a real-world end-to-end 3D teleconferencing scenario.

## 3.1 Sender Hardware

Our example implementation uses a single RGB camera (Logitech C920 Webcam HD Pro, 30 FPS, 1080p) and one computer with a GPU (NVIDIA GeForce RTX 3080) on the *Sender* side. The *Sender* side camera gives an RGB image per frame to the *Encoder* to perform face detection and head parameter extraction with neural networks executed on the GPU.

## 3.2 Head Model and Persistent Data Storage

Our *OpenTeleView* platform provides a Persistent Data Storage (PDS) model for data created by processes that are not run synchronously with the frame-by-frame streaming, such as a personalized head model; however, it can be accessed synchronously if desired; with the corresponding potential impact to performance.

Figure 2 illustrates one of the main use cases we envision: the *Encoder* is generic, trained once on a large dataset, and its parameters stored in the PDS and loaded at installation time; the *Decoder* is personalized (to the *Sender*), trained on the *Sender* side or external cloud, stored on the PDS, and network weights (354.4MB total size) are transmitted when a connection is made.

### 3.2.1 Head Model Predictor Training

In our illustration, the *Encoder* is generic as it is trained on a public data set with a range of people set rather than on a specific user. To show the modularity of our platform, we use either the self-supervised AutoLink [20] method or DECA [13], a pre-trained model for a generic 3D head model predictor. DECA is trained on over 21k subjects and 2 Million images from three publicly available datasets: VGGFACE2 [5], BUPT-Balancedface [49] and VoxCeleb2 [8]. The DECA model is learned in an analysis-by-synthesis way: input a 2D image $I$, encode the image to a latent code, decode this to synthesize a 2D image $I_r$, and minimize the difference between the synthesized image and the input.

### 3.2.2 Neural Renderer Training

The *Decoder* is personalized and we experiment with the architectures in [20] and [51]. The former is using a UNet neural network and the latter uses a more complex deferred renderer [45] with a caching mechanism to improve speed and runtime [51] (see Section 3.4). Both are trained using a short RGB video (approximately 5min) of the *Sender*. Videos are shot with a single fixed camera with the subject talking casually while performing small head motions, with a resolution of 1920x1080 at 60 FPS. The previously introduced *Encoder* models are used to obtain the driving motion from the *Sender*'s *talking head* video, specifically, AutoLink [20] extracts 2D keypoints and DECA [13] extracts the 3D head parameters. These head parameters are passed as inputs to the 2022 to reconstruct the encoded RGB image. Once training on this autoencoder objective is complete, the parameters of the neural renderer and the FLAME head shape parameters are stored in the PDS.

### 3.3 Encoder–Sender

The *Encoder* is a two-step process for each *Sender* frame to compute the Head Parameters: 1. finding the face of the *Sender* in the image and 2. using a pre-trained head model to compute the head parameters from the cropped *Sender*'s face image.

#### 3.3.1 Step 1: 2D Face Tracking

We extend a common approach to find a face bounding box around the *Sender's* face in the input image from a set of 68 2D face keypoints [18].

Previous methods [10, 11, 13, 15, 39, 47], run face detection, such as FAN [3], on every single frame, which is time-consuming and computationally heavy, leading to increased latency as 2D detection has to run before 3D reconstruction.

Instead, to achieve high FPS and low-latency head reconstruction on videos, we utilize

that there is a high temporal coherence of video data and propose to reuse the 2D face keypoints extracted from our reconstructed 3D head model of the previous frame to draw the face bounding box of the current frame. As this can lead to misalignment for fast motions, we further approximate the movement of the keypoints using a velocity estimate from the past two frames to extrapolate the position of current bounding box. A full face detection is performed when the bounding box displacement exceeds a threshold. This approach is robust to mispredictions and significantly reduces the time needed to detect and crop the face.

ices are projected into the image as $v = s\Pi(M_i) + t$, where $M_i \in R^3$ is a vertex in $M$, $\Pi \in R^{2 \times 3}$ is the orthographic 3D-2D projection matrix, and $s \in R$ and $t \in R^2$ denote isotropic scale and 2D translation respectively. The parameters $s$ and $t$ are summarized as an orthographic camera model $c$.

#### 3.3.2 Step 2: Extracting Head Parameters

With the cropped *Sender* face as input, a Head Parameter Extractor estimates fine-grained keypoint locations using a ResNet50 [19] followed by a fully connected layer to produce a latent code $e$, dependent on the used model, 2D keypoint locations $\mathbf{p} \in R^{32}$ and edge weights $\mathbf{w} \in 64$ or FLAME parameters, consisting of geometry $(\beta, \psi, \theta) \in R^{156}$, albedo coefficients $\alpha \in R^{50}$, camera matrix $c$, and lighting parameters $l$. This amounts to at most 2.5 KBytes/frame for encoding the 3D head model of a *Sender's* image. As only the time-varying pose information need to be sent every frame, the information sent for the 3D reconstruction is substantially less than what would be needed to send a whole 3D model of the *Sender*, greatly reducing network transmission time.

### 3.4 Decoder–Receiver

The *Decoder* is responsible for using the *Receiver*'s position $p$ and parameters $e$ predicted by the *Encoder* to reconstruct a view-dependent RGB image of the *Sender*. For the simpler 2D case, the decoder is a single network. Below we explain the 3D version that includes additional, view-dependent rendering steps.

There are two main steps in the process. First, the latent code $e$ is used to reconstruct the 3D head mesh of the *Sender*. Second, we use the personalized neural renderer to take the coarse 3D Head mesh, rotate it to the *Receiver*'s position, and generate a photo-realistic image of the *Sender*, view-dependent to appear on the view-dependent display.

#### 3.4.1 3D Neural Head Renderer

In the 3D setting, given the estimated FLAME parameters from the *Encoder*, the *Decoder* reconstructs the FLAME 3D head mesh using linear blend skinning (LBS) on parameters $e$. To ensure that head is consistently centered in the *Receiver*'s display, we rotate the mesh to the viewpoint $p$ and subtract the midpoint of vertices on each ear from all vertices on the mesh. One of our preliminary

baselines uses the coarse albedo parameters to texture and render the mesh. However, simple texture mapping is not photorealistic. Hence, we apply deferred neural rendering and first render the 3D mesh with UV coordinates as the texture. This UV map rendering then conditions the subsequent neural renderer along with a subset of the $e$ parameters. Lastly, because the FLAME parameters are predicted from a single image, we apply a small, one-sided box-filter to the pose ($\theta$) and the shape ($\beta$) parameters during online system evaluation.

#### 3.4.2 Cached 3D Neural Renderer

To accommodate for the the lower latency required for 3D teleconferencing, we use an optimized version [51] of the deferred neural renderer [45]. It is composed of two neural networks: a deep *caching* network that turns personalized neural textures to frame specific neural feature maps and a lightweight *warping* network that warps the feature maps cached from the previous frame.

The larger *caching network* can therefore be run sparingly, allowing to reduce the latency while minimally decreasing the visual quality of the generated image. On a multi-GPU machine, this method parallelizes and also increases the rendering frame-rate. Note that because this neural renderer is grounded with a 3D mesh, we are able to rotate the mesh (and thereby the UV map) to perform viewpoint-dependent rendering at inference time, based on the *Receiver*'s tracking data. Multiple viewpoints can be rendered for different display configurations, such as right/left perspectives for stereo.

### 3.5 Receiver Hardware

For our proof-of-concept implementation, the *Receiver* side hardware includes a spherical view-dependent display [55], a computer with a GPU, and a tracking system. In our current implementation, we explore the modularity of the platform by running the *Decoder* and Display processes on separate computers to illustrate that the display may be a self-contained system or the *Decoder* may be running using cloud services. However, they can also be run on a single computer. In Section 4.2, we analyze the timings of the different system components; thus, separating them allows us to consider this particular scenario.

#### 3.5.1 Spherical View-dependent Display + Computer

We use a large spherical view-dependent display [55], also known as a fish-tank virtual reality (FTVR) display. It uses a 24-inch plexiglass spherical screen with a mosaic of 4 registered mini projectors projecting through an 18-inch diameter hole at the bottom. This particular display is well suited for showing a view-dependent rendering of a *Sender*'s head because the spherical shape allows the *Receiver* to walk around the display and there are no seams. The mosaic of projectors provides a high-resolution, bright image. It has also been shown to be the most effective type of display for representing size and shape constancy which are important for human faces [57]. Lastly, the size of the sphere is large enough that a 1:1 aspect ratio is possible for human heads allowing for investigating whether the size of a 3D rendering of a speaker plays a role in perceived quality of presence. The display supports both view dependent and stereo depth cues. If such display is not available, our system also supports rendering to a flat screen.

#### 3.5.2 Tracking System

The tracking system provides *Receiver*'s position and viewing angle to the view-dependent display to achieve view-dependent rendering. The quality of view-dependent rendering is sensitive to errors in viewpoint tracking since it contributes significantly to the eye angular error pixels on a spherical view-dependent display [12, 56]. For our proof-of-concept implementation, we use OptiTrack (NaturalPoint Inc., Corvallis, OR) Prime-41 cameras to capture *Receiver*'s position

and orientation. This system uses retroreflective markers mounted on the Receiver's shutter glasses. The current tracking system has less than 0.2 mm of measurement error and the real-time streaming application connected with Unity has less than 10ms latency. The tracker data is used both by the *Decoder* and the Display Renderer (see 3.5.3). The *Decoder* uses *Receiver*'s position and orientation to render perspective dependent images for display.

### 3.5.3   Receiver Display Render

The rendering pipeline for the spherical display [12] is implemented in Unity (Unity Technologies, San Francisco, CA). It features a two-pass rendering approach: 1. render the image from a *Receiver*'s perspective, and 2. render the pixels on the output display. This separation enables the neural renderer to be trained display agnostic for planar frontal views while mapping to the desired display at runtime. For the spherical display, the second pass involves a mapping between 2D projector pixels to 3D surface positions on a non-planar surface. This warping transformation is achieved by sampling the 2D image texture in a shader program and using of the multiple-projector calibration matrix [55]. The same rendering pipeline also supports several different display modes, including mosaic display on the FTVR sphere, flatscreen display, and virtual display where you can freely move around the rendered objects in a virtual scene; thus, is versatile for researchers to experiment with different view-dependent display types.

We build on top of the two-pass rendering to further integrate the neural rendering into the pipeline by adding a rotating plane in the scene that is always normal facing the user and vertical. The neural renderer only requires the *Receiver*'s position and the thereby requested view is always up-right and onto a virtual planar image plane without distortion. To the user, they will always see the view corrected image based on their tracked position and display geometry. When they move around, this image and its orientation will be updated in real time by different aspects of the reconstructed talking head through neural rendering. This technique creates a sense of viewing 3D object while only rendering flat 2D images.

## 3.6   Tele-Communication Network

The goal of the telecommuncation network is to provide flexibility for where the computational resources are for each of the modules while at the same time providing an end-to-end infrastructure that mimics real-world conditions to support stress testing different modules used for 3D teleconferencing. Thus, we use a WebRTC backbone for communication with a ZMQ wrapper for each of the components in the platform. These are described next.

### 3.6.1   Internet backbone

We use the WebRTC protocol [40] using the *libdatachannel* [1] implementation to negotiate a direct peer-to-peer connection between the *Encoder-Sender* and the *Decoder-Receiver* over the internet. A WebRTC UDP configured data channel [21] facilitates the real-time transfer of 3D head parameters between the *Sender* and the *Receiver*. The 3D head parameters corresponding to a single frame are serialized using Protocol Buffers in order to be transmissible over the data channel. The UDP data channels are also used for data transfer between and Persistent Data Storage that is not local as well as the Tracker data to the *Decoder*. All the data channels are wrapped with a ZeroMQ [58] wrapper to provide a common interface for all the interprocess communication including support for different languages.

As the communication channels use UDP/IP with a ZeroMQ wrapper for all the communication interfaces, all the components of the end-to-end system can run on different machines as needed. Likewise, the interfaces between components have definitions for different language support enabling researchers to have flexibility in using C++, python or other languages to implement specific

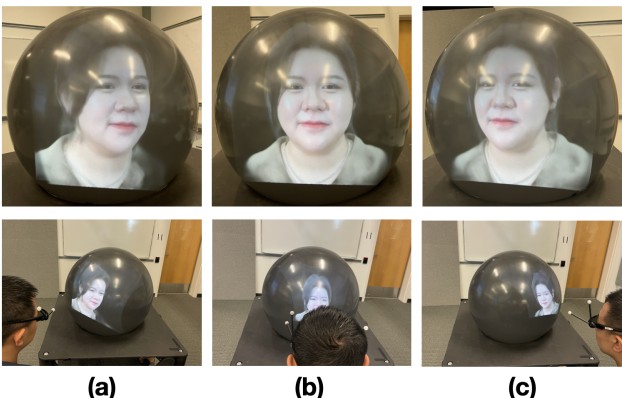

(a)                      (b)                      (c)

Figure 4: View-dependent rendering examples at different viewpoints: the first row shows the *Receiver*'s perspective and the second row shows the different positions of the *Receiver* by a fixed camera location. (a) Viewpoint at left of origin, seeing the right side of the *Sender*'s face (b) Viewpoint at origin, seeing the front side of the *Sender*'s face (c) Viewpoint at the right side of origin, seeing the left side of the *Sender*'s face.

algorithms. For example, in our current proof-of-concept implementation the networking is C++, the *Encoder* is implemented in Python/PyTorch, and the *Decoder* is Python/PyTorch.

Wrapping the communication channels supports the ability to send data structures seamlessly between different processes with different languages freeing the researcher to focus on using their preferred tool while the infrastructure takes care of the scaffolding needed to get the end-to-end system working for doing the analysis. Using this approach also ensures that components that are running on the same machine will do the data exchange locally.

## 4   MODULE EVALUATION IN *OpenTeleView*

We present results from analysis of each of our baseline modules when operating independently and as a part of the end-to-end system. The intent is to illustrate that performance analytics available within the end-to-end platform are effective to uncover inter-dependencies between components within the overall system and help to determine where bottlenecks in performance are coming from to guide algorithm development. The experiments reported here demonstrate the utility of testing modules in the *OpenTeleView* framework to address limitations otherwise unseen in isolated modules. We also show that our baseline end-to-end 3D teleconferencing implementation, along with the variations used for illustrating affects of changes to different modules, provide a good baseline for comparing future encoders/decoders/cameras and displays.

### 4.1   Module Evaluation Dataset

To illustrate evaluating performance of our individual modules, we recorded a 1920x1080 resolution, 60 FPS, stereo-view *talking head* dataset (*main* and *side* views) of one woman test subject. A second view is recorded to evaluate the *Decoder's* view-point dependent rendering capabilities. We include 5 sub-sequences in this dataset used for training, validation, and testing the *Decoder*, a sequence of fast-moving head motions for the *Encoder*, and a sequence for calibrating the cameras. We will make this dataset publicly available so others can evaluate their modules on the same data.

### 4.2   Baseline System Latency

Figure 5 shows the end-to-end live transmission system pipeline with FPS and latency of each corresponding component. The FPS results are generated by measuring the run time of each individual

| Pipeline Component | Camera A | Encoder | ZMQ | Internet | ZMQ | Decoder | ZMQ | Display | Total |
|---|---|---|---|---|---|---|---|---|---|
| FPS | 90 | **25** | >100 | >100 | >100 | 54 | >100 | >100 | **25** |
| Latency (ms) | 21 | 40 | 3 | 2 | 3 | 90 | 3 | 10 | **172** |

Figure 5: System Latency Breakdown: The blue coded parts are major system components, green coded parts are inter-transmission ZMQ, and the red parts are total FPS and latency. The total end-to-end latency with our computer hardware configuration is 172 msec at 25 fps. Additional latency due to the camera interface to the Open-TeleView components is dependent upon operating system drivers and are not included in these figures.

component. The theoretical latency is computed directly by taking the reciprocal of FPS. For comparison, we also estimated the perceptual latency by computing the time difference between the same movement of a real human and the rendered image on the display. To measure this, we use another high-speed camera to capture, in the same frame, the eye blink motion of a *Sender* talking and their image in the view-dependent display to calculate the time difference between blink motion. Using this approach we also take into account the OS and camera dependent delays to get an estimate of the overall system latency that would be in a real-system. With our particular camera hardware and OS, the perceptual latency is approximately 280ms, thus, non-encoder/decoder related elements contribute around 100ms of latency. The additional latency in the perceptual measurement comes from the time between eye-blink to the next frame capture (half a frame delay on average), asynchronous queue, and minimal smoothing applied to the estimated head parameters to mitigate jitter. Note that, due to the cached neural renderer, the perceptual novel-view-synthesis latency is much lower, at 35 ms, which facilitates a faithful VR experience even if the whole system communication is slower.

### 4.3 Velocity-based 2D Face Tracking

To evaluate the speed of the *Encoder* with our velocity-based 2D face tracking method, we independently test the *Encoder* using our recorded video of a subject moving their head quickly. The 1920x1080 at 60 FPS video contains 995 frames in total. Using our velocity method, the *Encoder* reruns the full face detection 58 times; without the velocity method, the *Encoder* reruns full face detection 676 times. Thus, it can be observed that our simple velocity method, that predicts the next frame's face location, can achieve a significant reduction in the number of times we have to rerun the time-costly face detection algorithm. From the perspective of the *OpenTeleView* platform affordance for this module, the timing information and the ability to switch between recorded video and live video feeds within the whole framework provides useful analytics to target timing bottlenecks to facilitate improving each module. In this case, we compared three different approaches that tradeoff face detection accuracy and computational load affecting latency.

### 4.4 Decoder Reconstruction Quality

To evaluate the quality of our displayed image, we independently test our *Decoder's* neural renderer on the withheld test sequence of our *talking head* dataset (*main* view). We are able to achieve a peak signal-to-noise ratio (PSNR) of 27.5 on the image from which the 3D head parameters have been estimated. Furthermore, we also test our models ability to perform view-dependent rendering by evaluating it on the second (*side*) view. This is done by taking the estimated 3D head parameters from the frontal recording, rotating those corresponding to the head pose based on the rotation matrix

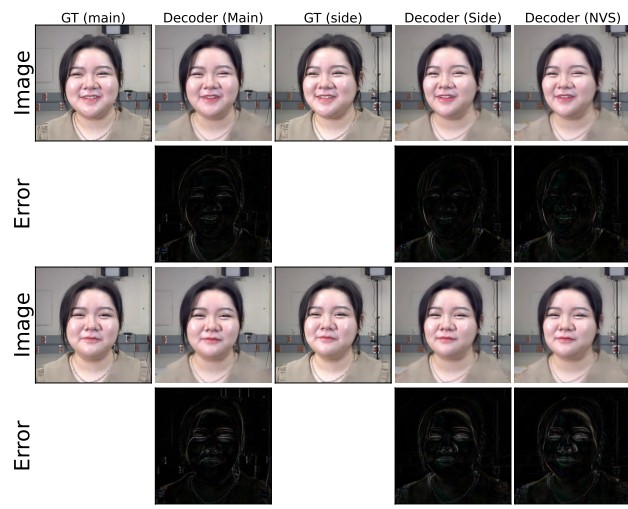

Figure 6: Comparison of our *Decoder* against the ground truth image for both *main* and *side* view examples. Examples where we perform novel-view synthesis (NVS) on the parameters from the *main* view are also shown.

between the *main* and *side* cameras. This is an especially difficult setting, for which our model was able to reconstruct the entire sequence with an average PSNR of 25.7. Note, when running our model on the estimated parameters from the *side* view, we are only able to achieve a PSNR of 26.7, showing that his side view is in general more difficult to reconstruct. Qualitative results and error maps can be seen in Figure 6.

### 4.5 *OpenTeleView* Modularity

To demonstrate the modularity of the proposed platform, we also experiment with the 2D encoder and decoder introduced in [20]. In this setting, we transfer the 2D keypoint locations **p** and their edge weights **w** obtained from the encoder. These are first rasterized into a coarse mesh, which is then lifted to a full image using a UNet. The latency for the encoder and decoder is respectively 4ms and 45ms. Figure 7 shows example images using this approach. It demonstrates that the platform can support entirely different encoder and decoder networks and corresponding parameterization (2D vs. 3D), without having to change the network communication or other parts of the framework.

### 4.6 *OpenTeleView* Integration

However, while each of our models is tested and developed using recorded videos and in isolation; when integrated, upstream delays in capturing and processing, such variable camera frame rates, leads to the degradation of downstream performance.

In the variable frame rate camera input, we observed that the velocity-based head tracking and warping-based neural renderer must compensate for increased differences between incoming frames. Further, the jitter associated with the incoming frames is not consistent, thus, a neural renderer may have variable time differences between frames, further challenging research that uses this approach. We illustrate how both modules are affected by changes in overall system framerate by subsampling frames in recorded videos and measure the performance versus the input framerate. These results are shown in Figure 8.

The ability for our *OpenTeleView* end-to-end platform to integrate different components easily enables isolation of each component's performance within real-world scenarios. Thus, in our example implementation, we illustrate that by switching in different encoder solutions and decoder solutions with both live feed and recorded video

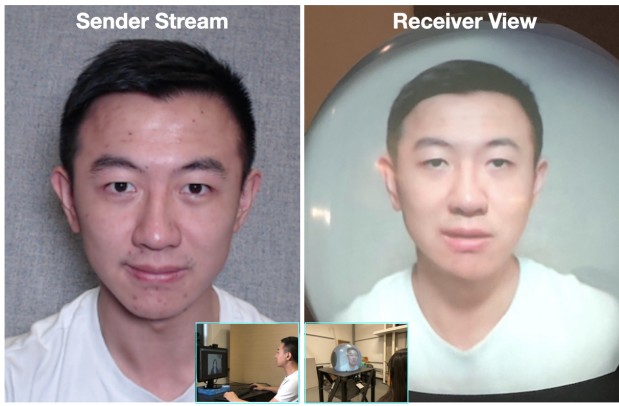

Figure 7: Example of modularity: we substituted using a 2D AutoLink method that conditions on 2D keypoints instead of a 3D mesh [20]. The encoder/decoder interface makes this a simple operation so that researchers can swap different approaches to compare performance in real-world like end-to-end teleconferencing.

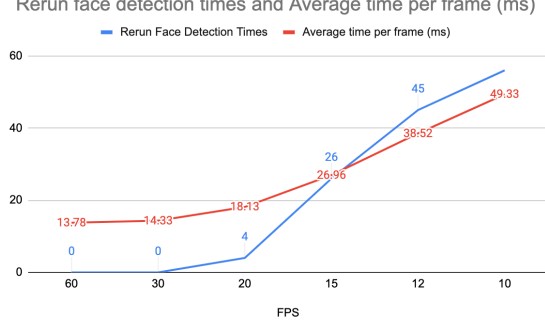

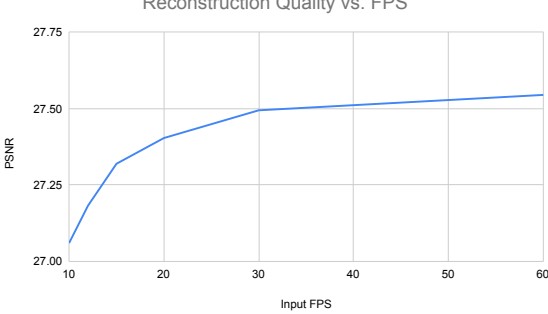

Figure 8: Reduced input video/image frame rates negatively impact the performance of both the velocity-based head tracking and the neural rendering.

feeds, careful performance analysis, the strengths and weaknesses of each component are identified along with the inter-dependencies amongst the components. Thus, our *OpenTeleView* platform fills a significant gap in assessing different computer vision approaches to the encoder and decoder methods that are usually assessed only in isolation on pre-recorded data sets. Hence, our contribution enables *apples-to-apples* comparisons of different algorithms for 3D teleconferencing.

## 5 LIMITATIONS AND FUTURE WORK

The focus of this paper is on the *OpenTeleView* platform rather than the specifics of the baseline encoder/decoder pair we implemented to provide a particular baseline. In that context, even though our platform has most of the major modules implemented for end-to-end 3D teleconferencing, there are some components which we leave for future work. These include: additional **analytics** such as temporal and spatial jitter measurements; additional **baseline use-cases** such as multicamera and mobile displays; **symmetric communication**, **multicast abilities**; embedded, synchronized **audio support** rather than out-of-band audio; and, **parameterized input control** so that the input video stream characteristics can be easily adjusted to simulate different real-world camera input statistics.

## 6 CONCLUSION

We created the *OpenTeleView* platform along with two baseline implementation that illustrates how the end-to-end 3D teleconferencing can work and future research on separate modules can be analyzed. The baseline 3D implementation provides a medium fidelity teleconferencing experience using modifications of existing techniques available in the literature. The second method uses a much simpler 2D representation to illustrate the support for modularity and flexibility of the encoder/decoder to support a range of approaches researchers may investigate. The platform is intended to use off-the-shelf components for computational, camera and display hardware along with an internet-based communication infrastructure so that it is accessible to a large range of researchers. This approach enables research on specific approaches to encode the input video and decode it to provide view-dependent rendering needed for 3D teleconferencing to be tested and analysed in a common end-to-end platform. By doing so, research contributions on specific modules can be tested in real-world scenarios to facilitate constant innovations in 3D teleconferencing technology to lead the way for establishing this new form of remote communication.

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
