# OpenReview forum: "OpenTeleView: An Open 3D Teleconferencing Research Platform"
_graphicsinterface.org/Graphics_Interface/2023/Conference_SD — Submitted to GI 2023 - second deadline_

### Official Review · Reviewer_yRa2 · 2023-04-21
**Final project report, not scientific paper**

**Rating:** 5
**Confidence:** 4

**Review:**

The claimed contributions of the work are:

"1 describe [an] open platform for real-time end-to-end 3D teleconferencing
2 demonstrate integrating state-of-the art modules and improving them based on an analysis of current bottlenecks
3 [provide] open source implementation and benchmark"

The primary contribution is #3. However, the supplement does not contain the code, there is no link to an (anonymized) github or other repository, and the paper doesn't provide even the module API in an appendix. So, I cannot evaluate the primary contribution. The related work section does not distinguish the modular system from previous work, except to note that previous work is proprietary. (The related work section does explain the difference between the specific example modules and previous work, but not the system).

What remains is a systems paper describing the high level design of a system and low-level design of some implementation modules, but without significant means of evaluating the system or distinction from previous work. The systems sounds reasonable and I think is worth presenting in some context. However, this is more of a whitepaper or technical report than a falsifiable scientific systems paper or engineering paper with demonstrated novelty.

---

### Official Review · Reviewer_m5mT · 2023-04-24
**Promising, but missing important aspects**

**Rating:** 4
**Confidence:** 3

**Review:**

This paper introduces OpenTeleView, an end-to-end 3D teleconferencing platform. OpenTeleView aims to provide a common platform for the development, testing, and analysis of 3D teleconferencing system components. The authors present two baseline implementations to illustrate the adaptability and modularity of the platform. In addition, they assess the performance of each module when operating independently and as part of the overall system.

The paper focuses on the OpenTeleView platform rather than the encoder/decoder pairs used as a baseline. This is understandable given the primary purpose of the platform, but it limits the discussion of the techniques used in the baseline implementations. It would have been interesting to compare the encoding/decoding part to other SOTAs, such as Instant Neural Graphics Primitives (INGP), which can reconstruct the radiance field in less than 2 seconds. In fact, one could argue that changing the platform's architecture to allow transmitting a small number of images (instead of pose and face parameters) and employing INGP-like methods on the decoder side is a faster/more efficient solution.

The authors include an extensive evaluation of the performance of individual modules and the end-to-end system. This analysis includes the dataset used, baseline system latency, velocity-based 2D face tracking, decoder reconstruction quality, and the platform's modularity and integration capabilities.

Several important components, such as temporal and spatial jitter measurements, additional baseline use cases, symmetric communication, multicast capabilities, embedded synchronized audio support, and parameterized input control, are left for future work. At least the synchronized audio support should have been included in the first prototype.

The paper does not compare to existing platforms or solutions in the domain of 3D teleconferencing.

To summarize, OpenTeleView is a promising platform for end-to-end 3D teleconferencing that addresses a gap in the research community. Flexibility, modularity, and exhaustive evaluation are notable strengths of the platform. However, the paper's limited scope, unaddressed components, lack of comparison to existing solutions, and absence of an evaluation of user experience are significant weaknesses in my opinion.

Also, there are some major parsing/editing issues in Section 3.3.1.

---

### Official Review · Reviewer_RSqY · 2023-04-25
**Interesting approach, but not new**

**Rating:** 3
**Confidence:** 5

**Review:**

I do not see this fitting into the “research” category of papers, this is an assembly of other work, and should be in the “systems” category (if one exists) – I tried to look for the categories available, but I cannot find what are available – definitely not research.
Treating it as a systems paper:
The encoder (compression) is not very clear at all, the only thing I can find is that the encoder is presenting a 2.5K/frame set of data – it seems that the texture is transmitted once (and then the expression updated). The expression update is not good, so it’s a proof of concept, not a good tech demonstration. The quality is low and therefore the big question is how does this compare to the data rate and quality of MPEG Multiview video (which takes advantage of redundancy between camera views) – in addition, cameras are very cheap now. VVC (latest version of video) has a very low data rate for very high quality; 2.5K/frame is 60Kbytes/s, which is 480Kbits/s which is quite a high data rate by comparison to today’s video standards. I am not sure how the PSNR calculation is achieved, given that PSNR is a pixel to pixel comparison (for video) and from what I can tell you are unlikely to achieve this, in addition PSNR values of 25.7db and 26.7db are considered very low quality (and then 1db difference is not very significant or comparable by traditional video conferencing methods) – but as I said, I am not sure how you are comparing in the first place (more information would help). All this to say that with these results we are unlikely to be changing to this method in the future.
The requirement to train on 5m of video is also cumbersome, but within reason acceptable if the data is stored properly.
Some of the references are useless, for example ZMQ is only a github repository which explains no more than the paper explains (which is pretty much that it’s a wrapper) … this needs to be clarified in the paper.
The velocity-based face tracking is either poorly explained or not very significant; change the text to whatever the method is and how it best represents your approach.
The information in Figure 8 doesn’t really tell me anything important and I don’t see the “apples-to-apples comparison” that you mention… this should have been assessed using participants and subjective testing.
This is an interesting first step, but the system is complicated and the presented work doesn’t really tell us anything that is useful. The video is useful and interesting, but I would have preferred to have seen more examples of different people, rather than lots of transitions.
Overall, I would say both the system and the paper still need some work.